# Revealing the high variability on nonconserved core and mobile elements of *Austropuccinia psidii* and other rust mitochondrial genomes

**Jaqueline Raquel de Almeida**[1], **Diego Mauricio Riaño Pachón**[2], **Livia Maria Franceschini**[1], **Isaneli Batista dos Santos**[1], **Jessica Aparecida Ferrarezi**[1], **Pedro Avelino Maia de Andrade**[1], **Claudia Barros Monteiro-Vitorello**[1], **Carlos Alberto Labate**[1], **Maria Carolina Quecine**[1] *

1 Department of Genetics, "Luiz de Queiroz" College of Agriculture (ESALQ), University of São Paulo, Piracicaba, São Paulo, Brazil, 2 Center for Nuclear Energy in Agriculture (CENA), University of São Paulo, Piracicaba, São Paulo, Brazil

* mquecine@usp.br

**Data Availability Statement:** The complete sequence of the mitochondrial genome of A. psidii

## Abstract

Mitochondrial genomes are highly conserved in many fungal groups, and they can help characterize the phylogenetic relationships and evolutionary biology of plant pathogenic fungi. Rust fungi are among the most devastating diseases for economically important crops around the world. Here, we report the complete sequence and annotation of the mitochondrial genome of *Austropuccinia psidii* (syn. *Puccinia psidii*), the causal agent of myrtle rust. We performed a phylogenomic analysis including the complete mitochondrial sequences from other rust fungi. The genome composed of 93.299 bp has 73 predicted genes, 33 of which encoded nonconserved proteins (ncORFs), representing almost 45% of all predicted genes. *A. psidii* mtDNA is one of the largest rust mtDNA sequenced to date, most likely due to the abundance of ncORFs. Among them, 33% were within intronic regions of diverse intron groups. Mobile genetic elements invading intron sequences may have played significant roles in size but not shaping of the rust mitochondrial genome structure. The mtDNAs from rust fungi are highly syntenic. Phylogenetic inferences with 14 concatenated mitochondrial proteins encoded by the core genes placed *A. psidii* according to phylogenetic analysis based on 18S rDNA. Interestingly, *cox1*, the gene with the greatest number of introns, provided phylogenies not congruent with the core set. For the first time, we identified the proteins encoded by three *A. psidii* ncORFs using proteomics analyses. Also, the *orf208* encoded a transmembrane protein repressed during *in vitro* morphogenesis. To the best of our knowledge, we presented the first report of a complete mtDNA sequence of a member of the family Sphaerophragmiacea.

has been deposited in GenBank (accession number MN018834).

**Funding:** This study was supported by Fundação de Amparo à Pesquisa do Estado de São Paulo FAPESP (Grant 2014/16804-4). We thank FAPESP for the fellowship award to JRA (2016/16868-8) and LMF (2015/14344-9). We also thank Coordenação de Aperfeiçoamento de Pessoal de Nível Superior (CAPES) for the fellowship award to IBS and Conselho Nacional de Desenvolvimento Científico e Tecnológico (CNPq) to JAF and PAMA.

**Competing interests:** The authors have declared that no competing interests exist.

## Introduction

Rust fungi, classified as the most devastating diseases worldwide, are widely distributed in nature [1]. *Austropuccinia psidii* [2,3] is an obligate biotrophic plant pathogen that is the causal agent of myrtle rust. This pathogen has evolved specialized structures, such as haustoria, formed within the host tissue to efficiently acquire nutrients and suppress host defenses [4,5].

Myrtle rust, first described as occurring on leaves of *Psidium guajava* L. (*Psidium pomiferum* L.) (Myrtaceae) in Brazil [2], can now infect various species within the genus Eucalyptus [6,7]. Quecine *et al*. [8] suggested that it is most likely due to its high genetic variability within populations [8]. In South America, rust is a significant threat to *Eucalyptus grandis*, one of the most cultivated species. Moreover, it is becoming significant to Myrtle species in its origin center Australia. To date, 358 native species from 49 genera identified in Australia were susceptible to rust [9]. There are several efforts to improve knowledge about the biology of *A. psidii*, such as the sequencing of its nuclear genome [10–13]; proteomic of urediniospores [14]; effects of cuticular waxes on fungal germination [15]; confirmation of its sexual life cycle [12,16]; among others. Recently, taxonomic studies have led to the reclassification of *A. psidii*. A maximum-likelihood phylogenetic analysis using the sequences of the nuclear ribosomal RNA genes suggested that *A. psidii* does not belong to Puccinia but should be within the new genus *Austropuccinia* of Pucciniales in the redefined family Sphaerophragmiaceae [3].

Fungal mitochondrial genomes are typically small, circular, and double-stranded DNA molecules. These genomes usually harbor at least 14 protein-coding conserved genes, namely, apocytochrome b (*cob*); three subunits of the cytochrome c oxidase (*cox1*, *cox2*, *cox3*); seven subunits of the NADH subunits (*nad1*, *nad2*, *nad3*, *nad4*, *nad4L*, *nad5*, and *nad6*); three subunits of the ATP synthase (*atp6*, *atp8*, *atp9*); as well as the large and small ribosomal RNA (rRNA) subunits (*rnl* and *rns*); a set of tRNAs genes; and the RNA subunit of the mitochondrial RNase P (*rnpB*). The genomes also present a variable number of groups-I and -II introns that may bear homing endonuclease genes (HEGs) with LAGLIDADG or GIY-YIG motifs [17–19]. HEGs are selfish genetic mobile elements that encode site-specific-sequence-tolerant DNA endonucleases. The catalytic activity of HEGs promotes their propagation by introducing DNA double-strand breaks (DSBs) into alleles lacking the endonuclease-coding sequence and by the subsequent repair of these DSBs via homologous recombination using the endonuclease-containing allele as a template [20]. mtDNA also harbors numerous repetitive sequences, besides introns and plasmids, known as mtDNA instability agents, generating variability within fungal mitochondrial genomes [21,22]. According to Kolesnikova *et al*. [23], the variation in mtDNA size in four different *Armillaria* species is due to variable numbers of mobile genetic elements, introns, and plasmid-related sequences. Most *Armillaria* introns contained open reading frames (ORFs) related to homing endonucleases of the LAGLIDADG and GIY-YIG families.

Mitochondrial genomes evolve independently of and faster than the nuclear genome [24]. It is often useful as a valuable source of information to study systematics and evolutionary biology in eukaryotes where insufficient phylogenetic signals have accumulated in nuclear genes [17,25]. For instance, Song *et al*. [26] resolved some incongruences of Dothideomycetes phylogeny using the mitochondrial genome of many phytopathogens belonging to this Class [26]. The availability of mitochondrial genome sequences provides valuable information about genome organization and enables evaluating structural rearrangements using comparative studies [27]. The high rate of polymorphism frequently found within introns or intergenic regions of well-conserved mitochondrial genes makes these sequences useful for genetic diversity studies, both among and within populations [28–31].

In addition to its role in energy production and other essential cellular processes, mitochondria may also participate in fungal pathogenesis [25]. For instance, mitochondrial β-

oxidation plays an essential role in vegetative growth, conidiation, appressorial morphogenesis, and pathogenesis progression in *M. oryzae* [31]. The methylation of the mitochondrial genome is an epigenetic mechanism affecting the adaptation and pathogenicity of *Candida albicans* [32]. Furthermore, mitochondrial metabolic functions are targets for pathogen control [33,34].

Based on the critical role of mitochondria to phytopathogenic fungi and the need to better understand the myrtle rust causal agent, we sequenced, assembled, and annotated the *A. psidii* mitochondrial genome. We thoroughly characterized the genome's gene content and organization, codon usage, and repetitive elements. We also explored the evolutionary dynamics of the mitochondrial genomes of rust fungi by a comparative mtDNA focused on mobile element analysis. Finally, throughproteomic approaches, we identified three previously hypothetical mitochondrial proteins unique to *A. psidii*.

## Materials and methods

### DNA sequencing and mitochondrial genome assembly

*A. psidii* monopustular isolate MF-1 was previously obtained from *E. grandis* [35]. The high-molecular-weight DNA of MF-1 was obtained from urediniospores using the DNeasy Plant mini kit (Qiagen). A NanoVue spectrophotometer quantified the extracted DNA, and the quality was checked by agarose gel electrophoresis. Total DNA was used to generate libraries for 454 pyrosequencing (Roche), PacBio SMRT sequencing on an RSII instrument (Pacific Biosciences), and sequencing by synthesis on a MiSeq Instrument (Illumina).

We obtained *A. psidii* mtDNA sequences by mining reads from 454 and MiSeq platforms using mitochondrial reference genomes.

Complete mitochondrial genome sequences from all available species of representative rust fungi (Pucciniales) were obtained from the NCBI database and Puccinia's comparative genomics projects of the Broad Institute (Table 1).

Mining was performed using the Mirabait program in the MIRA package using the MITObim approach. All steps were performed using modules of the MIRA sequence assembler software in "mapping mode" to map reads to a reference and create new reference sequences; and an in silico-baiting module, which is used to extract reads that precisely match a given reference across a number of n k-mers of length k (defaults n = 1 and k = 31) from the entire set of reads [36]. Finally, mitochondrial reference genomes were used to mapping single molecule

**Table 1. Complete mitochondrial genome of rust pathogens used in present work.**

| Reference organisms | Data source * | Pathogen's characteristics |
|---|---|---|
| *Phakopsora meibomiae* Puerto_Rico | NCBI (NC_014352.1) | Causal agent of American rust. It occurs mainly in soybean crops. |
| *Phakopsora pachyrhizi* Taiwan_72–1 | NCBI (NC_014344.1) | Causal agent of Asian soybean rust. It occurs mainly in soybean crops. |
| *Puccinia graminis* f. sp. *tritici* | *Puccinia*—Group Database Broad Institute | Causal agent of stem rust. It occurs in wheat, barley, rye, triticale and some other species of Poaceae (Gramineae). |
| *Puccinia striiformis* PST-78 | *Puccinia*—Group Database Broad Institute | Causal agent of wheat yellow rust, which occurs in crops of wheat and barley. |
| *Puccinia triticina* 1-1BBBD-race-1 | *Puccinia*—Group Database Broad Institute | Causal agent of wheat leaf rust. |
| *Moniliophthora perniciosa* | NCBI (NC_005927.1) | Causal agent of "witches' broom disease" of the cocoa tree. |

*\*Puccinia*—Group Database Broad Institute http://www.broadinstitute.org/annotation/genome/puccinia_group/MultiHome.html.

NCBI - http://www.ncbi.nlm.nih.gov/.

sequencing reads from SMRT platform using basic local aligment using BlasR packge (https://github.com/PacificBiosciences/blasr) developed by PacificBioscicnce [37].

The assembly of the *A. psidii* mitochondrial genome from reads and subreads of MiSeq, 454, and SMRT platforms were performed using SPAdes v. 3.7 (http://cab.spbu.ru/files/release3.7.0/manual.html) with automatic coverage cutoff [38]. QUAST v. 4.0 (http://bioinf.spbau.ru/quast) was used to compute assembly metrics and validate assembly quality [39,40]. All computational analyses were performed according to software tutorials.

## Annotation of the mitochondrial genome

*A. psidii* mitochondrial genome was annotated using the default parameters of MFannot (https://github.com/BFL-lab/Mfannot) [41] and GeSeq (https://chlorobox.mpimp-golm.mpg.de/geseq.html) [42]. The annotation was adjusted manually using the BLASTx tool available in the NCBI, restricting the similarity search to the "Pucciniales" order (taxid:5258). The Genome Vx tool (http://wolfe.ucd.ie/GenomeVx/) was used to plot each gene's position and orientation from the mitochondrial genome of *A. psidii*.

For the identification of the transfer RNAs (tRNAs), the tRNAscan-SE software (http://lowelab.ucsc.edu/tRNAscan-SE/) was used with default parameters [43]. The GC content analysis of the mitochondrial genome was performed with program Genomics % GC Content Calculator (http://www.sciencebuddies.org/science-fair-projects/project_ideas/Genom_GC_Calculator.shtml).

The complete sequence of the mitochondrial genome of *A. psidii* was deposited in GenBank (accession number MN018834).

## Comparative and phylogenetic analysis of rust mtDNAs

Comparative and phylogenetic analyses were performed among the rust mtDNA from *A. psidii*, *P. graminis f. sp. tritici*, *P. triticina*, *P. striiformis*, *P. meibomiae*, and *P. pachyrizi* (Table 1). The mitochondrial genome sequence of *Moniliophtora perniciosa* (NC_005927.1) was used as an outgroup [44]. The mitochondrial genomes of *P. graminis f. sp. tritici*, *P. triticina*, *and P. striiformis*, *P. meibomiae*, *P. pachyrhizi*, *M. perniciosa* were reannotated using MFannot and GeSeq and manually verified to avoid errors in comparative and phylogenetic analyses. The nonconserved ORF (ncORFs) from all mtDNA rust fungi were annotated by BLAST analysis. We defined as ncORFs all predicted ORFs that did not belong to the mitochondrial core protein-encoding genes. We also used the tRNAscan-SE software and Genomics % GC Content Calculator to identify tRNAs and GC% content in the rust mtDNAs.

The fourteen conserved mitochondrial protein sequences were used for phylogenetic analyses: cytochrome c oxidase (cob, cox1, cox2, cox3), ATP synthase subunits (atp6, atp8, atp9), and NADH dehydrogenase subunits (nad1, nad2, nad3, nad4, nad4L, nad5, and nad6). Protein sequences were aligned with MUSCLE implemented in MEGA X [45]. Poorly aligned amino acid regions were removed using TrimAl (http://trimal.cgenomics.org/) [46]. Proper evolutionary models for phylogenetic inference were computed with MrModeltest v. 2.3 (https://github.com/nylander/MrModeltest2) [47] using the Bayesian Information Criterion (BIC). Specifically, mtREV24 was determined as the best model for *cob*, LG+I+F for *cox1*, cpREV for *cox2*; WAG+I+F for *cox3*; mtREV24+G for *atp6*; mtREV24 for *atp8*; mtREV24+I for atp9; mtREV24 for *nad1*; WAG+I+F for *nad2*; mtREV24+G for *nad3*; JTT+I+F for *nad4*, cpREV for *nad4L*; WAG+G+I+F for *nad5*; JTT+G+F for *nad6*. The 14 protein sequences of each pathogen were concatenated using Mesquite software v. 3.2 (https://www.mesquiteproject.org/) [48], and Bayesian phylogenetic inference was carried out with MrBayes v. 3.2.7 (https://nbisweden.github.io/MrBayes/download.html). The Bayesian analysis included two separate

runs of 1 x 107 generations, sampled every 1000 generations, and 25% of the initial generations.

In addition, the phylogenetic relationship of *A. psidii* within rust fungi was evaluated using the 18S rDNA and *cox1* genes. Multiple sequence alignments were generated with MUSCLE [49], trimmed with TrimAl, and selected the best evolutionary model described above (LG+I +F for cox1, T92+G+I for 18S rDNA). The phylogenetic tree was inferred as previously mentioned using MrBayes.

## Proteomics analysis

Data processing, protein identification, and relative quantitative analyses of proteomics data were performed using the raw data previously obtained by Quecine *et al.* [14] using the ProteinLynx Global Server (PLGS- v 2.5.1). The reanalysis of the *A. psidii* proteome was performed to validate the presence of hypothetical proteins in the *A. psidii* mitochondrial genome. The processing parameters were set, according to Quecine *et al.* [14]. Briefly, to identify the proteins, the intensities of the spectra were calculated by the stoichiometric method, according to the internal standard, the sequence of rabbit phosphorylase (Uniprot entry: P00489), by MSE analysis [50] and normalized using the PLGS auto normalization function. Protein identifications were obtained with the embedded ion accounting algorithm of PLGS software searching into the *A. psidii* mitochondrial proteins appended in the internal standard. All protein hits were identified with confidence of >95%.

A database was created based on the predicted proteins obtained from the manual annotation of the mitochondrial genome of *A. psidii*, as well as on mitochondrial protein sequences of *P. graminis* f. sp. *tritici*, *P. triticina*, *P. striiformis* (Broad Institute's Puccinia—Group Database). Protein identification was obtained with the embedded ion accounting algorithm of PLGS software. After PLGS 2.5.1 analysis, the data were manually inspected, and the parameters obtained using MassPivot v. 101 were included: (i) the average amount (fmol) of protein, (ii) the average score of proteins, and (iii) the average amount of matched peptides to each protein.

For the hypothetical proteins found on the proteomics data, we performed additional in silico analyses. The peptide sequences were obtained by EMBOSS Transeq (https://www.ebi.ac. uk/Tools/st/emboss_transeq/) and then the conserved motifs evaluated by MOTIF, Genome-Net of the Kyoto University Bioinformatics Center (http://www.genome.jp/tools/motif/).

## RT-qPCR analysis

To validate the proteomic analysis, gene expression of the three ncORFs, *orf174*, *orf205*, *orf208*, identified by mass spectrometry, was evaluated by RT-qPCR during the fungal in vitro morphogenesis as described below. The set of primers for ncORFs was generated using Oligo-Perfect™ Designer software (http://tools.lifetechnologies.com/content). The presence of dimers and hairpins was verified using Oligo Analysis Tool software (http://www.operon.com/tools/oligo-analysistool.aspx). The genes of beta-tubulin and elongation factor were used as references [51] (Table 2).

*In vitro* morphogenesis experiment: *A. psidii* urediniospores from the monopustular isolate MF-1 were inoculated on dialysis membranes on agar-water medium (8 g L$^{-1}$) amended with 0.5% of olive oil [52,53]. Dialysis membranes were sampled at each interval: zero hours after inoculation (h.a.i.), 6 h.a.i. (absence of germination), 12 h.a.i. (germination tubes formation), and 24 h.a.i. (appressoria formation) [35]. Total RNA of four biological replicates for each time was isolated using a spin column procedure employing a Spectrum Plant Total RNA Extraction Kit (Sigma-Aldrich). RNA isolation procedure followed the manufacturer's

**Table 2. Primers used in this study.**

| Target gene | Primer | Sequence 5'- 3' | Reference |
|---|---|---|---|
| *beta-tubulin* | BTub1 | GGACTCTGTTTTAGATGTCGTC | Bini *et al.* 2017 |
| | BTub3 | TTGATGGACTGATAGGGTAGCG | Bini *et al.* 2017 |
| *elongation factor* | EF5 | CAGTTATGGAAGTTTGAAACTCC | Bini *et al.* 2017 |
| | EF2 | GACAATAAGCTGTCGAACACCAAGG | Bini *et al.* 2017 |
| *orf174* | Po174F | GGCACACGACCTCTGTACCT | This study |
| | Po174R | TTCACAAGATGCAGGCTCAC | This study |
| *orf205* | Po205F | TGCAGAGAAGGATGCACAAC | This study |
| | Po205R | TCAAAAGCATGAACCATTCG | This study |
| *orf208* | Po208F | GAAGGTAAGCGGGAGGGTA | This study |
| | Po208R | TTCTACCCCGTTCTATTCTATCC | This study |

protocol, including on-column DNase digestion (Sigma-Aldrich). RNA concentration and A260/A280 ratios were measured for each sample by a NanoVue Spectrophotometer (GE Healthcare). Quantified RNA samples were stored at -80˚C. One μg of each RNA sample was reverse-transcribed to cDNA using a High Capacity cDNA Reverse Transcription Kit (Applied Biosystems, Foster City, USA), employing random hexamer primers. Reverse-transcribed samples were stored at −20˚C. The RT-qPCR reactions were prepared in a final volume of 12.5 μl containing 6.25 μl of Platinum® SYBR® Green qPCR SuperMix-UDG, 10 pmol of each primer set, 2.5 μl cDNA (1:20) and 1.75 μl ultrapure water. The samples were amplified in an iCycler IQ® Real-Time PCR Detection System (Bio-Rad) with the following conditions: 95˚C for 15 min (1x); 95˚C for 15 sec, 58˚C for 30 sec, 72˚C for 30 sec (45x); and 71 cycles of 60–95˚C with a progressive increase of 0.5˚C per cycle (melting). Each sample was analyzed in two technical duplicates.

Reaction efficiencies were analyzed using the LinRegPCR program (version 11.0), and the relative expression values were calculated by the Pfaffl method [54] using the REST software (Relative Expression Software Tool) [55]. The Pairwise Fixed Reallocation Randomization Test calculated the differential expression with 1000 bootstrap iterations, and the analyzed intervals were 6–0 h.a.i., 12–0 h.a.i. and 24–0 h.a.i.

## Results

### Mitochondrial genome of *A. psidii*

The mt genome assembly resulted in two contigs: 62,940 bp and 30,490 bp. By using CAP3 [56], the contigs were joined with a small overlapping region (approximately 70 bp), confirmed by PCR, resulting in a single contig of 93.299 bp (Fig 1). The *A. psidii* mitochondrial genome sequence presented a GC content of 37.39%, very similar to other published rust mtDNA.

### Mitochondrial genome annotation

The MFannot predicted 73 different ORFs. Among them, the 14 mitochondrial core protein-coding genes were identified (Table 3). The results showed that 23 tRNA genes and all other 14 core genes transcribe from the same DNA strand clockwise (Fig 1). Details of *A. psidii* MF-1 core proteins were presented in S1 Table.

Besides, we identified 33 nonconserved ORFs (ncORFs). The ncORFs represent approximately 45% of the total genes found in this genome. We identified intronic ncORFs in 11 introns of conserved genes (one in the *cob*, one in the *cox2*, and 9 in the *cox1* genes). These

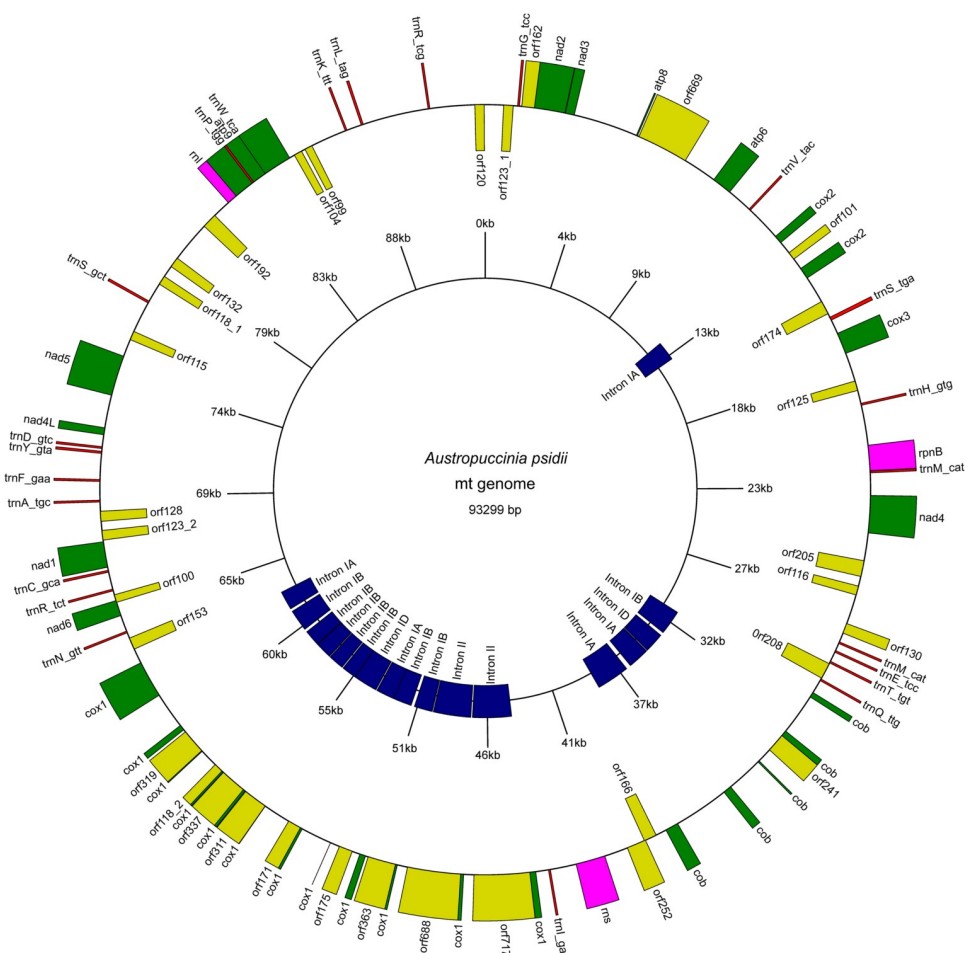

**Fig 1. The mitochondrial genome of *Austropuccinia psidii* MF-1.** The mitochondrial view was obtained by the Genome Vx tool (http://wolfe.ucd.ie/GenomeVx/).

ncORFs encode for a ribosomal protein, LAGLIDADG endonucleases, and reverse transcriptases (RT) (Table 3). Some of the ncORFs have alternative start codons. In contrast to the conserved core genes, the ncORFs were found on both strands (S2 Table). In other rust mtDNA, only LAGLIDADG endonucleases were detected as HEG (S3 Table).

The mitochondrial genome was screened for codon usage, and the genes were analyzed for their start and stop codons (S1 and S2 Tables). Among the predicted genes, the 'AUG' was the initiation codon in all 14 conserved core genes. However, in ncORFs, the start codons 'UAU', 'GAA', 'AGU', 'UGA', 'CAA', 'GGU', 'GCC', and 'ACG' were found. The most frequent stop codon was 'UAA' (Fig 2).

## Rust mtDNA comparative phylogenetic analysis

Among the six rust mtDNA evaluated, *A. psidii* had the largest number of genes and mtDNA size, followed by *Puccinia* spp. (77,600 bp on average). The *Phakospsora* spp. had the smallest rust mitochondrial genome (32,172 bp on average). The GC content of *A. psidii* mtDNA did not differ from other rust mtDNAs. The *Phakospsora* spp. mtDNA lacked the *rpnB* gene. The organization of core genes was conserved among the rust mitochondrial genomes (Fig 3A and

**Table 3. Predicted genes of *Autropuccinia psidii* MF-1 mitochondrial genome.**

| Predicted genes | Encoded protein | Interesting features |
|---|---|---|
| **Respiratory chain proteins** | | |
| **Complex I** | | |
| *nad1* | NADH dehydrogenase subunit 1 | contiguous and in phase with *orf162* |
| *nad2* | NADH dehydrogenase subunit 2 | |
| *nad3* | NADH dehydrogenase subunit 3 | |
| *nad4* | NADH dehydrogenase subunit 4 | |
| *nad4L* | NADH dehydrogenase subunit 4L | |
| *nad5* | NADH- dehydrogenase subunit 5 | |
| *nad6* | NADH- dehydrogenase subunit 6 | |
| **Complex III** | | |
| *cob* | Cytochrome b | 4 introns |
| | | I2—group = ID |
| | | I3 and I4 -group = IA(5') |
| **Complex IV** | | |
| *cox1* | Cytochrome c oxidase subunit 1 | 11 introns |
| | | group II–I1 and I2 |
| | | group = IA–I5 and I12 |
| | | group = IB–I3, I4, I7, I8, |
| | | I9, I10 and I11 |
| | | group = ID–I6 |
| *cox2* | Cytochrome c oxidase subunit 2 | 1 intron |
| *cox3* | Cytochrome c oxidase subunit 3 | alternative ATG start pos 17117 |
| **Complex V** | | |
| *atp6* | ATP synthase subunit a | |
| *atp8* | ATP synthase subunit b | 1 intron |
| *atp9* | ATP synthase subunit 9 | alternative ATG start pos 85321 |
| **rRNA** | | |
| *Rns* | - | |
| *Rnl* | - | |
| **Other proteins** | | |
| *rpn*B | recombination-promoting nuclease RpnB | |
| **ncORFs** | | |
| *orf99* | hypothetical protein | Intergenic ncORF |
| *orf100* | hypothetical protein | Intergenic ncORF; in opposite strand of nad6 |
| *orf101* | LAGLIDADG endonuclease | Intronic ncORF (*cox2*-I1) |
| *orf104* | hypothetical protein | Intergenic ncORF; |
| | | in same sequence oposite strand of ATP9 |
| *orf115* | hypothetical protein | Intergenic ncORF |
| *orf116* | hypothetical protein | Intergenic ncORF; TTG upstream: 27503 |
| *orf118_1* | hypothetical protein | Intergenic ncORF |
| *orf118_2* | LAGLIDADG endonuclease | Intronic ncORF (*cox1*-I9) First aa- Tyr |
| *orf120* | hypothetical protein | Intergenic ncORF |
| *orf123_1* | hypothetical protein | Intergenic ncORF |
| *orf123_2* | hypothetical protein | Intergenic ncORF |
| *orf125* | hypothetical protein | Intergenic ncORF |
| *orf128* | hypothetical protein | Intergenic ncORF |

(*Continued*)

**Table 3.** (Continued)

| Predicted genes | Encoded protein | Interesting features |
|---|---|---|
| *orf130* | hypothetical protein | Intergenic ncORF |
| *orf132* | hypothetical protein | Intergenic ncORF |
| *orf153* | hypothetical protein | Intergenic ncORF; TTG upstream: 64208 |
| *orf162* | hypothetical protein | *nad2* gene contínuos and in phase with *orf162* |
| *orf166* | DEAD/DEAH box helicase dominion | Intergenic ncORF |
| *orf171* | LAGLIDADG endonuclease | Intronic ncORF (*cox1*-I9); First aa- Glu; |
| *orf174* | hypothetical protein | Intergenic ncORF |
| *orf175* | LAGLIDADG endonuclease | Intronic ncORF (*cox1*-I4) |
| *orf192* | hypothetical protein | Intergenic ncORF |
| *orf205* | hypothetical protein | Intergenic ncORF; TTG upstream: 26703 |
| *orf208* | hypothetical protein | Intergenic ncORF |
| *orf241* | LAGLIDADG endonuclease | Intronic ncORF (*cob*-I2) First aa–Ser |
| *orf252* | ribosomal protein S3 | Intergenic ncORF |
| *orf311* | LAGLIDADG endonuclease | Intronic ncORF (*cox1*-I7); Codon alternative para UGA para Trp |
| *orf319* | LAGLIDADG endonuclease | Intronic ncORF (*cox1*-I10) First aa–Ile |
| *orf337* | LAGLIDADG endonuclease | Intronic ncORF (*cox1*-I8) First aa–Gln |
| *orf363* | LAGLIDADG endonuclease | Intronic ncORF (*cox1*-I3) First aa- Gly |
| *orf669* | reverse transcriptase | Intronic ncORF (*atp8* I1) |
| *orf688* | reverse transcriptase | Intronic ncORF (*cox1*-I2) First aa- Gly |
| *orf717* | reverse transcriptase | Intronic ncORF(*cox1*-I1) First aa- Thr |

3B). However, the total size of intronic and intergenic regions among them was highly variable. *A. psidii* has the largest amount of ncORF sequences (Fig 3C) and intronic sequences.

Genes *cox1* and *cob* presented introns in mtDNA of all rust pathogens (Table 4). Almost all of the introns detected in six of all the genes carried some HEG. The number of LAGLIDADG endonucleases encoded in intronic ncORF ranged from three in the soybean rust (*P. pachyrhizi*) to nine in *A. psidii*. Compared with other rust pathogens, *A. psidii* showed a higher

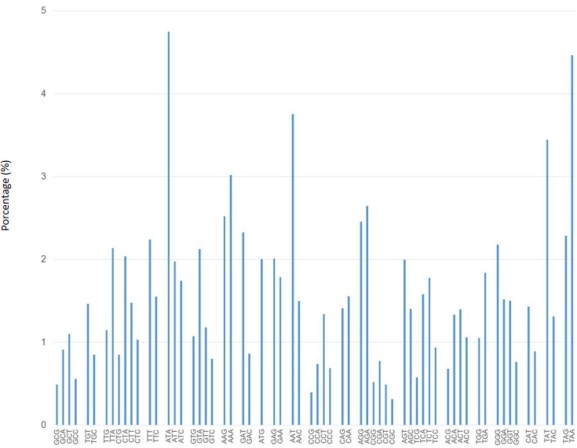

**Fig 2. Column diagram of the *Austropuccinia psidii* mtDNA codon usage.** The diagram represents the codons (x-axis) and percentages of their occurrence (y-axis) in the *A. psidii* mitochondrial genome.

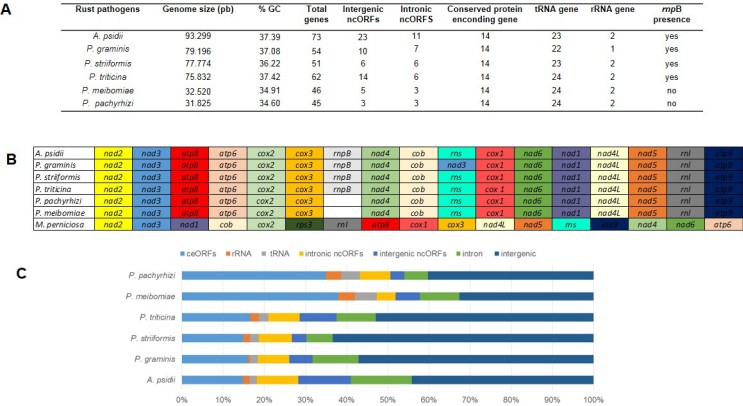

**Fig 3. Comparison of the mitochondrial genome from *Austropuccinia psidii* and other rust.** (A) General comparison of mitochondrial genome content among *A. psidii* and *P. graminis*, *P. striiformis*, *P. triticina*, *P. meibomiae* and *P. pachyrhizi*, (B) Organization of core genes among the rust mitochondrial genomes, (C) Proportions of ceORFs, rRNA, tRNA, intronic and intergenic ncORFs, intron and intergenic content in the rust mitochondrial genomes.

diversity of HEGs and intron groups: nine LAGLIDADG endonucleases, three reverse transcriptases, and six intron group IA, eight IB, two ID, and two II (S4 and S5 Tables).

Rust mtDNAs encode a similar number of tRNA genes. Only *A. psidii* has two tRNAs for glutamic acid (Glu) and a unique anticodon for lysine (Lys) (Table 5).

We compared the phylogeny using all 14 core proteins encoded by the mtDNA with the one of the *cox*1. The results showed that the multigene approach supported *A. psidii* as a sister clade to *Puccinia* spp. However, using *cox1*, the result was different, *A. psidii* clustered with *Phakospsora* spp., although with low branch support (Fig 4A and 4B). The 18S rDNA-based phylogenetic analysis corroborated the clustering of *A. psidii* as a sister clade of *Puccinia* spp. (S1 Fig).

## Proteomic and RT-qPCR analysis

Of the 33 ncORFs found in *A. psidii* mtDNA, we identified three of them in a previously generated proteomic dataset [14]: *orf174*, *orf205*, and *orf208*. Only *orf174* was present in other rust fungi, such as *P. graminis* (S6 Table). *Orf205* and *orf208* were unique to mtDNA of *A. psidii* (S6 Table). Using the software MOTIF, we found a conserved domain in *orf174*, identified as belonging to DNA topoisomerase I superfamily cl27598. *Orf205* has two conserved domains: one similar to the peroxidase family2 and another described as DUF2070. The *orf208* has the DUF2070 domain, as well (S7 Table).

We performed RT-qPCR analysis to confirm the expression of *orf174*, *orf205*, and *orf208* during fungal *in vitro* morphogenesis. Only the expression of *orf208* was detected. We

**Table 4. Number of introns in conserved protein-coding genes in mtDNA of rust pathogens.**

| Rust pathogens | atp8 | cob | cox1 | cox2 | nad4 | nad5 | Total |
|---|---|---|---|---|---|---|---|
| *A. psidii* | 1 | 5 | 12 | 1 | | | 19 |
| *P. meibomiae* | | 1 | 4 | | | | 5 |
| *P. pachyrhizi* | | 1 | 4 | | | | 5 |
| *P. graminis* | | 2 | 5 | 1 | 1 | 2 | 11 |
| *P. striiformis* | | 2 | 6 | 1 | 1 | 1 | 11 |
| *P. triticina* | | 2 | 7 | 1 | | 1 | 11 |

**Table 5. tRNAs present in the rust pathogens mtDNA.**

| Amino Acid | *P. meibomiae* | *P. pachyrhizi* | *P. graminis* | *P. striiforms* | *P. triticinia* | *A. psidii* |
|---|---|---|---|---|---|---|
| | Anticodon | Anticodon | Anticodon | Anticodon | Anticodon | Anticodon |
| Ala | TGC | TGC | TGC | TGC | TGC | TGC |
| Arg | TCG | TCG | TCG | TCG | TCG | TCG |
| | TCT | TCT | TCT | TCT | TCT | TCT |
| Asn | GTT | GTT | GTT | GTT | GTT | GTT |
| Asp | GTC | GTC | GTC | GTC | GTC | GTC |
| Cys | GCA | GCA | GCA | GCA | GCA | GCA |
| Gln | TTG | TTG | TTG | - | TTG | TTG |
| Glu | TTC | TTC | TTC | TTC | TTC | TTC |
| | - | - | - | - | - | TCC |
| Gly | TCC | TCC | TCC | TCC | TCC | - |
| His | GTG | GTG | GTG | GTG | GTG | GTG |
| Ile | GAT | GAT | - | GAT | GAT | GAT |
| Leu | TAG | TAG | TAG | TAG | TAG | TAG |
| Lys | TTT | TTT | TTT | TTT | TTT | TTT |
| | CTT | CTT | CTT | CTT | CTT | - |
| Met | CAT | CAT | CAT | CAT | CAT | CAT |
| | CAT | CAT | CAT | CAT | CAT | CAT |
| Phe | GAA | GAA | GAA | GAA | GAA | GAA |
| Pro | TGG | TGG | TGG | TGG | TGG | TGG |
| Ser | GCT | GCT | GCT | GCT | GCT | GCT |
| | TGA | TGA | TGA | TGA | TGA | TGA |
| Sup | TCA | TCA | TCA | TCA | TCA | TCA |
| Thr | TGT | TGT | TGT | TGT | TGT | TGT |
| Tyr | GTA | GTA | - | GTA | GTA | GTA |
| Val | TAC | TAC | TAC | TAC | TAC | TAC |
| | **24** | **24** | **22** | **23** | **24** | **23** |

observed a gradual downregulation of the expression during the development of germinative tubes and appressorium formation (Fig 5).

## Discussion

Compared with other mtDNAs rust pathogens used in the present study, *A. psidii* had one of the largest ones. Fungal mitochondrial genomes are highly variable in size, ranging from 12 kb in the mycoparasite *Rozella allomyces* [57] to 235.8 kb of the fungus *Rhizoctonia solani* [58]. The mtDNA size variability may occur among organisms from the same species. The mtDNA from *P. striiformis* f. sp. *tritici* ranged 102,521 [59], approximately 25% bigger than the mtDNA from *P. striiformis* PST-78, used in the present study. Several factors contribute to size variations, including the proliferation of noncoding sequences such as short tandem repeats, gene duplication followed by inactivation, intron expansion, and incorporation of foreign sequences from different sources [25,58,60]. According to Medina *et al.* [25], within Dikarya in general, Basidiomycetes mtDNA is highly variable in gene order compared to Ascomycetes. Furthermore, while in Basidiomycetes, mtDNAs have genes usually encoded on both strands, in Ascomycetes, they are encoded in only one. Interestingly in *A. psidii* and other evaluated rust mtDNAs, the core genes are in the same strand.

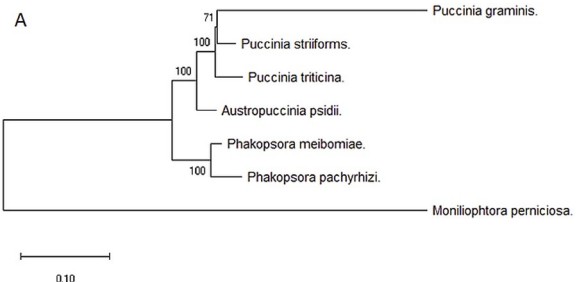

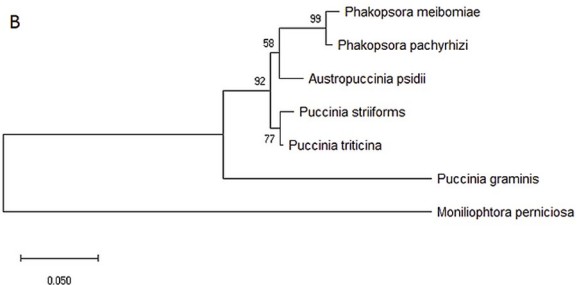

**Fig 4. Maximum likelihood phylogenetic tree of rust mtDNA.** The phylogeny analyses were based on the whole genome (A) and *cox1* (B) sequences. The sequences were aligned using the MUSCLE The statistical method Maximum Likelihood (bootstrap test with 1000 repetitions) and the Hasegawa—Kishino—Yano model were used for the phylogenetic tree construction. The numbers above tree nodes represent the bootstrap support values. *M. perniciosa* was used as an out-group.

The large size of *A. psidii* mtDNA is partially associated with the abundance of ncORF. ncORFs are frequently reported in fungal mitochondrial genomes [61–65], but their origin and function are still unknown. For instance, the variable sizes of the mitochondrial genomes

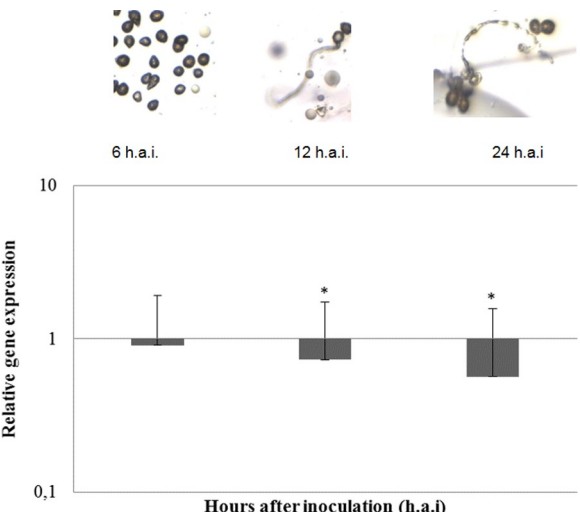

**Fig 5. RT-qPCR analysis of *orf208* gene.** Expression analysis of *orf208* from *A. psidii* MF-1 in three different times of *in vitro* fungal development. Expression values were normalized with beta-tubulin and elongation factor as a housekeeping reference Pfaffl [54]. Values represent the expression ratio of the *orf208* gene. Bars represent the mean for three replicates, and error bars show the standard error of the mean. Asterisks indicate values that differ significantly (P_0.05) between 0 and 6, 12 or 24 h after inoculation as determined by the method of Pfaffl et al. [55].

of *Colletotrichum* species are due, in part, to the presence or absence of ncORFs and intronic sequences [66]. The large mtDNA size of *Phlebia radiata* (156 kbp) also harbors a large number of introns and long intergenic regions [63].

*A. psidii* had the largest number of ncORFs in intronic regions among the mtDNAs assessed in this work. Many of the ncORFs were related to HEGS, primarily LAGLIDADG endonucleases. Queiroz *et al.* [66] observed that *C. lindemuthianum* contains just one intronic ncORF encoding a LAGLIDADG endonuclease. However, more than one intronic ncORF encoding LAGLIDADG endonucleases were found in rust mtDNA. HEGs can expand mtDNA size, cause genome rearrangements, gene duplications, and import exogenic nucleotide sequences through horizontal gene transfer (HGT) [67,68]. HEGs may also be involved in the spread of group I introns between distant species [69,70].

Concerning the HEGs found in rust mtDNA, we identified only LAGLIDADG encoding genes. These sequences are self-splicing and play relevant roles in processes associated with genome evolution [71]. Many group I intron-encoded LAGLIDADG proteins function as maturases assisting in RNA splicing [72–75]. This activity described in fungi such as *Saccharomyces cerevisiae*, *Saccharomyces capensis*, *Aspergillus nidulans* [73,75,76], suggests that endonuclease and maturase activities are close in function and evolution to LAGLIDADG proteins encoded by group-I introns in rust fungi [77]. GIY-YIG ORFs have been reported in introns of fungal mitochondria [77,78]. However, no GIY-YIG endonucleases were found in rust mtDNA.

Mobile elements, including HEGs, play a crucial role in the expansion of fungal mitochondrial genomes. We observed a relationship between the genome size and total mobile elements hosted in intronic sequences in rust fungi. The number of introns is highly variable among mitochondrial genomes; for example, *Fusarium graminearum* has 34 group-I introns [79], whereas *Mycosphaerella graminicola* has no introns [80]. Ambrosio *et al.* [81] revealed that 48.7% of *C. cacaofunesta* mtDNA was composed of introns. The number of intron groups in rust fungi mtDNA ranged between 3–18 to *P. meibomiae* and *A. psidii*, respectively. There is a lack of available complete mtDNA sequence of Pucciniales. Thus, our research may bring a significant contribution to the comprehension and phylogeny of this group. In our studies using mtDNA from Pucciniales, phylogenetic analyses revealed that according to core genes, *A. psidii* was confidently a sister clade of *Puccinia* spp. This result is in agreement with the 18S rDNA phylogenetic analysis. Similarly, Zhang *et al.* [82] observed a congruency between nuclear ribosomal RNA and mitochondrial protein-based trees to *Cordyceps militaris*. Our phylogeny data demonstrated that mitochondrial core genes are an alternative for determining phylogenetic relationships among rust fungi, as shown in other species of fungi and other organisms [83–85].

We need broader taxon sampling to include all the phylogenetic diversity of the group and achieve a robust phylogeny and evolutionary trajectory, including the mobile element. The mobile genetic elements and features such as the number of introns per gene and similar positions were more similar between *A. psidii*—*Puccinia* spp. than that of *A. psidii*–*Phakopsora*.

Interestingly, comparing only *cox1* from complete mtDNA of the six Pucciniales, *A. psidii* was closer to *Phakopsora* spp, although with notably low branch support. This data supports the fact that since the degree of conservation and organization of genes may vary according to the group studied, a preliminary analysis is essential to select "a priori" reliable phylogenetic markers [25]. Mitochondrial genes, such as *cox1*, were widely used for barcoding many groups of organisms, although with less identification power in the fungal kingdom due to the rapid evolution of their mt genomes [27]. Wang *et al.* [86] demonstrated that the frequent heteroplasmy and recombination in the mitochondrial genomes of *Thelephora ganbajun* resulted in two types of introns in different sites of *cox1* with varying frequencies among the isolates.

Allelic association analyses of the observed mitochondrial polymorphic nucleotide in *cox1* sites suggested that mtDNA recombination is frequent in natural populations of this fungus. Li *et al.* [87] assembled the mtDNAs of *Pleurotus citrinopileatus* and *Pleurotus platypus* and observed thirteen classes of introns (Pcls) within the *cox1* gene. The number and class of Pcls varied among different *Pleurotus* species, indicating that the introns in *cox1* directed the mitochondrial genome rearrangements. Only concatenated mitochondrial protein sequences were suitable as molecular markers for phylogenetic analysis of *Pleurotus* spp.

We identified three out of the 33 ncORFs in our previous proteomic datasets [14]. The only conserved region of these proteins was the DUF2070 domains, with TM-regions found in *orf208* and *orf205*. According to Tang *et al.* [88], the mitochondrial membrane protein FgLetm1, containing DUF2070, regulates mitochondrial integrity, production of endogenous reactive oxygen species and mycotoxin biosynthesis in *Fusarium graminearum*. The authors obtained ΔFgLetm1 mutant that significantly reduced endogenous ROS levels, decreased mycotoxin deoxynivalenol biosynthesis, and attenuated virulence *in planta*. Thus, we suggest that *orf208* and *orf205* encode a transmembrane protein that may be related to fungal pathogenicity. According to the RT-qPCR results, the only transcribed gene was *orf208*. The gene was down-regulated during the fungal morphogenesis.

. Nine of the 33 ncORFs identified in the *A. psidii* mtDNA presented alternative start codons. We strongly believe that they are not pseudogenes. It is known that mtDNA and other plastid genomes are composed of a significant number of genes with alternative codons, most of them related with mobile elements [89,90]. The nine ncORFs (*orf118_2*, *orf171*, *orf241*, *orf311*, *orf319*, *orf337*, *orf363*, *orf688*, *orf717)* with alternative start codons were associated with mobile elements that may be related to the evolution of the mtDNA in progress [77,91]. More assays using different stimuliand assays carried out *in planta* should be performed to validate the ncORFs presence encoding protein to help understandthe function of these ncORFs in mtDNA *A. psidii*. We also showed the first experimental evidence of three new mitochondrial proteins exclusive of *A. psidii*. Furthermore, the functional characterization of these proteins and their association with particular mitochondrial pathways or during host interaction is a valuable tool to elucidate important biological features of myrtle rust disease.

## Supporting information

**S1 Fig. Maximum likelihood phylogenetic tree of rust pathogens based on 18S rDNA partial sequences obtained from the GenBank nucleotide sequence database.** The accession numbers are in parentheses. The sequences were aligned using the MUSCLE method. For the phylogenetic tree construction, the statistical methods Maximum Likelihood, the Bootstrap method test with 1000 repetitions, and the Hasegawa—Kishino—Yano model were performed. The numbers above tree nodes represent the bootstrap support values. *M. perniciosa* was used as an out-group.
(DOCX)

**S1 Table. Conserved gene features of the *Austropuccinia psidii* MF-1 mitochondrial genome.**
(DOCX)

**S2 Table. Nonconserved ORFs (ncORFs) features of the *Austropuccinia psidii* MF-1 mitochondrial genome.**
(DOCX)

**S3 Table. Nonconserved ORFs (ncORFs) features in mtDNA rust pathogens.**
(DOCX)

**S4 Table. Number of LAGLIDADG endonucleases and intron types present in introns of six genes gene in mtDNA of rust pathogens.**
(DOCX)

**S5 Table. Features of introns characterized in mtDNA of rust pathogens.**
(DOCX)

**S6 Table. Nonconserved ORFs (ncORFs) in mtDNA *Austropuccinia psidii* MF-1 shared by other rust pathogens.**
(DOCX)

**S7 Table. Conserved domain in unknown function proteins found in *Austropuccinia psidii*.**
(DOCX)

## Acknowledgments

We thank Dr. Thais Regiani for supporting the proteomic analysis. We are grateful for the efforts of Dr. Andressa Peres Bini in developing the previous RT-qPCR protocol analyses.

## Author Contributions

**Conceptualization:** Jaqueline Raquel de Almeida, Diego Mauricio Riaño Pachón, Claudia Barros Monteiro-Vitorello, Maria Carolina Quecine.

**Data curation:** Jaqueline Raquel de Almeida, Diego Mauricio Riaño Pachón, Livia Maria Franceschini, Pedro Avelino Maia de Andrade.

**Formal analysis:** Livia Maria Franceschini, Isaneli Batista dos Santos, Pedro Avelino Maia de Andrade.

**Funding acquisition:** Carlos Alberto Labate, Maria Carolina Quecine.

**Investigation:** Jaqueline Raquel de Almeida, Diego Mauricio Riaño Pachón, Isaneli Batista dos Santos, Jessica Aparecida Ferrarezi, Pedro Avelino Maia de Andrade, Maria Carolina Quecine.

**Methodology:** Jaqueline Raquel de Almeida, Diego Mauricio Riaño Pachón, Livia Maria Franceschini, Isaneli Batista dos Santos, Jessica Aparecida Ferrarezi, Pedro Avelino Maia de Andrade, Maria Carolina Quecine.

**Resources:** Maria Carolina Quecine.

**Software:** Diego Mauricio Riaño Pachón, Livia Maria Franceschini, Pedro Avelino Maia de Andrade.

**Supervision:** Maria Carolina Quecine.

**Writing – original draft:** Jaqueline Raquel de Almeida, Maria Carolina Quecine.

**Writing – review & editing:** Jaqueline Raquel de Almeida, Diego Mauricio Riaño Pachón, Livia Maria Franceschini, Isaneli Batista dos Santos, Jessica Aparecida Ferrarezi, Pedro Avelino Maia de Andrade, Claudia Barros Monteiro-Vitorello, Carlos Alberto Labate, Maria Carolina Quecine.

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
