## [Decision Letter · Decision Letter 0]

25 Sep 2020

PONE-D-20-26480

Revealing the high variability on nonconserved core and mobile elements of Austropuccinia psidii and other rust mitochondrial genomes

PLOS ONE

Dear Dr. Quecine,

Thank you for submitting your manuscript to PLOS ONE. After careful consideration, we feel that it has merit but does not fully meet PLOS ONE’s publication criteria as it currently stands. Therefore, we invite you to submit a revised version of the manuscript that addresses the points raised during the review process.

Please address all questions and comments raised by the reviewers. With respect to point 3 raised by reviewer 2, I realize that obtaining additional RNA-seq or proteomics data is outside of the the scope of this study, especially during the current coronavirus pandemic, but please make sure that you have used all available data or otherwise indicate where additional data might be needed to distinguish true genes from pseudogenes.

We look forward to receiving your revised manuscript.

Kind regards,

Minou Nowrousian

Academic Editor

PLOS ONE

Journal Requirements:

Reviewers' comments:

Reviewer's Responses to Questions

**Comments to the Author**

1. Is the manuscript technically sound, and do the data support the conclusions?

Reviewer #1: Yes

Reviewer #2: Partly

2. Has the statistical analysis been performed appropriately and rigorously? 

Reviewer #1: N/A

Reviewer #2: Yes

3. Have the authors made all data underlying the findings in their manuscript fully available?

Reviewer #1: Yes

Reviewer #2: Yes

4. Is the manuscript presented in an intelligible fashion and written in standard English?

Reviewer #1: Yes

Reviewer #2: No

5. Review Comments to the Author

Reviewer #1: This is a well-prepared manuscript describing the mitochondrial genome of A. psidii, a rust fungus. It is of interest to the plant pathogenic fungi research community. I have several minor comments.

1. line 22, upon first mentioning, it is unclear what is "most species".

2. line 125, the sequence mining analysis is not described in detail enough that it can be reproduced.

3. line 153, same problem with the "Blast" analysis, which should be spelled as "BLAST".

4. Several another computational analysis throughout the manuscript were also not described in detail enough.

5. ClustalW is a relatively old tool. I think newer alignment tools like MAFFT, or MUSCLE, could help the authors achieve better alignment quality.

6. line 415, this section in discussion reads more like a reiteration of results. The authors should speculate a bit more as to why a trans-membrane protein is down-regulated during morphogenesis. What potential role it might play?

Reviewer #2: In the present manuscript, the mitochondrial genome of the rust fungus Austropuccinia psidii was sequenced and annotated. Besides the mitochondrial core genes, a large number of non-conserved predicted genes were identified in this genome, some of which have an alternative start codon. Three of the latter were confirmed by analysis of an existing proteome dataset. Only for one of them, expression was detectable at the RNA level. It is downregulated during infectious development. In parallel, the mitochondrial genes also enable phylogenetic comparisons which overall aligns well with the 28S phylogeny.

While additional mitochondrial genome sequences for rust fungi are of value to the community and the descriptive analysis is well-done, in the present state the manuscript appears premature. I suppose, the dataset originates from whole genome sequencing, and the mitochondrial genome was analysed separately. The manuscript would benefit from rewriting so that one or a few main findings are put into focus that shape the storyline.

I was left with these open questions:

1. What is the main point that is really special/novel about this mitochondrial genome?

2. The choice of genomes for comparison is not well explained. For example, two additional genomes from Puccinia striiformis f. sp. tritici (Pst) CY32 and P. recondita f. sp. tritici (Pt) HnZU18-3 were published last year. Why are these excluded? In particular since the former is larger than the A. psidii genome sequenced here.

3. Out of 33 ncORFs only 3 have support in proteome data, and of these only one is found at the RNA level. I would like to see more data supporting the predicted ORFs, e.g. from RNAseq or additional proteome analysis. In particular the ones with alternative start codons: could these be pseudogenes?

4. The homing endonucleases, in particular the GIY type, are highlighted. is there any evidence that they shaped evolution in this family?

Additional points:

Overall: the English language needs to be corrected by a native speaker. There are minor mistakes throughout the entire manuscript.

Line 23 and throughout the manuscript: In my opinion, it should be rust fungi, not fungi rusts. But please verify with a native speaker.

Line 113 and 236-239: It seems a lot to use 3 different sequencing methods for a mitochondrial genome. Please explain the contribution of each of the datasets to the final genome assembly.

Table 1: Please explain the choice of mitochondrial genomes chosen for comparison in the table heading. Also, readability might be improved by re-formatting the table.

Line 236: Here, I am missing an explanation of the quality parameters. What makes this assembly best?

Line 239: When reading this, I wondered if 37 % GC is typical. Later this information is given, but this is one example that illustrates why I would like the storyline to improve.

Table 3 and table 4: Both tables are rather lengthy. I would suggest to move them to the supplements are excel files, so that the reader can filter e.g. by position. For the main text, extract the most relevant features and make corresponding tables that only show, what is also discussed in the text. Orientation for example could also be included in fig. 1.

Line 324: orf414 should be orf 208

Line 326 and 330: Please adjust formatting of the reference.

Line 356: Reference is missing

Discussion: The more specific parts of the discussion could be moved to the results section to streamline the manuscript. In my view this would make the result section more interesting to read, and the discussion could focus on the highlight and the open questions of mitochondrial genome research.

6. PLOS authors have the option to publish the peer review history of their article (what does this mean?). If published, this will include your full peer review and any attached files.

Reviewer #1: No

Reviewer #2: No

---

## [Author Response · Author response to Decision Letter 0]

9 Feb 2021

February 4th, 2021

LETTER RESPONSE

Dear Academic Editor 

Dr. Minou Nowrousian

PlosOne

This is a summary of the revision made in the manuscript entitled “Revealing the high variability on nonconserved core and mobile elements of Austropuccinia psidii and other rust mitochondrial genomes” (PONE-D-20-26480). We are grateful for the contributions made by the editor and reviewers. We tried to answer all the questions and we have also accepted all the suggestions made by them. The response to editor and reviewers’ comments is enclosed. 

Best regards,

Maria Carolina Quecine 

 

Response to Editor

We are grateful for the opportunity to resubmit our manuscript. We believe that all reviewer’s comments collaborated with the manuscript’s improvement. 

• Concerning your observation “With respect to point 3 raised by reviewer 2, I realize that obtaining additional RNA-seq or proteomics data is outside of the the scope of this study, especially during the current coronavirus pandemic, but please make sure that you have used all available data or otherwise indicate where additional data might be needed to distinguish true genes from pseudogenes”, we confirm a non-available data from A. psidii to distinguish genes from pseudogenes. Thank you for your understanding about the limitation to obtain new data under the coronavirus pandemic. We also believe that it is outside of the scope of this study, however we carefully investigated about pseudogenes and A. psidii mtDNA as described below and with more detail in response to reviewer 2.

We agree that some ncORFs may be pseudogenes, however this possibility is not strong and we appointed some information to support our view. Firstly, we revised the alternative start and stop codons that were found in the ncORFs from A. psidii. A new annotation was made in MFannot and revised the result manually. In first draft we found 12 ncORFs harboring alternative star or stop codons (orf118_2,, orf123_1, orf162, orf171, orf174, orf241, orf311, orf319, orf337, orf363, orf688, orf717). From the new annotation, we observed that ncORFs, orf123_1, orf162 and orf174 were wrong characterized. The correct stop codon in orf123_1 is UAG. Similar mistake were observed in orf162 and orf174 in these ncORFs the stop codon are UAG and UAA respectively. All these mistakes were corrected in the manuscript.

Moreover, it is known that the presence of alternative codons is an indicative of pseudogenes. However, it is also known that mtDNA and other plastids genomes are composed with a great amount of genes with alternative codons, most of them related with mobile elements with functionality as the maturases (Zoschke et al. 2010, Keren et al. 2009). The 9 ncORFs (orf118_2, orf171, orf241, orf311, orf319, orf337, orf363, orf688, orf717) with alternative start codons that were found in mtDNA from A. psidii are associated with mobile elements. Most pseudogenes arise as copies of functional genes, either directly by DNA duplication or indirectly by reverse transcription of an mRNA transcript (Zheng et al. 2007). To verify this hypothesis we blasted our sequences with the own A. psidii mtDNA and with mtDNA from other rust mtDNA including CY32 and HnZU18-3 organisms. None similarity was found, except with the previously annotated mobile elements. Thus, the review of ncORFs with alternative start codons support previously studies that describe alternative start codons with mobile elements related with the evolution of the mtDNA in progress (Wenli et al. 2008, Chevalier and Stoddard 2001). The discussion about this question was added in the discussion section and we hope that have improved the manuscript. 

References: 

Chevalier BS, Stoddard BL. Homing endonucleases: structural and functional insight into the catalysts of intron/intein mobility. Nucleic Acids Res. 2001. 29(18): 3757–3774.doi: 10.1093/nar/29.18.3757

Keren I, Bezawork-Geleta A, Kolton M, Maayan I, Belausov E, Levy M, et al. AtnMat2, a nuclear-encoded maturase required for splicing of group-II introns in Arabidopsis mitochondria. RNA. 2009;15:2299–2311. doi: 10.1261/rna.1776409. 

Wenli Jia, Paul G. Higgs, Codon Usage in Mitochondrial Genomes: Distinguishing Context-Dependent Mutation from Translational Selection, Molecular Biology and Evolution, Volume 25, Issue 2, February 2008, Pages 339–351, https://doi.org/10.1093/molbev/msm259

Zheng D, Frankish A, Baertsch R, Kapranov P, Reymond A, Choo SW, et al. Pseudogenes in the ENCODE regions: Consensus annotation, analysis of transcription, and evolution. Genome Res. 2007;17:839–851. doi: 10.1101/gr.5586307.

Zoschke R, Nakamura M, Liere K, Sugiura M, Börner T, Schmitz-Linneweber C. An organellar maturase associates with multiple group II introns. Natl Acad Sci U S A. 2010;107:3245–3250. doi: 10.1073/pnas.0909400107. 

 

Response to Reviewer 1

Thank you for considering our manuscript well prepared. We really appreciated all comments that certainly will improve our manuscript. The answers concerning the reviewer’s comments are listed below:

• Concerning the comment “Line 22, upon first mentioning, it is unclear what is "most species"”, we changed the phrase using “many fungal groups”.

• We agree with the observation “line 125, the sequence mining analysis is not described in detail enough that it can be reproduced”. Thus, we rewrote the methodology “Mining was performed using the Mirabait program in the MIRA package using the MITObim approach. All steps are performed using modules of the MIRA sequence assembler software, which is used in mapping mode to map reads to a reference and create new reference sequences; and an in silico-baiting module, which is used to extract reads that precisely match a given reference across a number of n k-mers of length k (defaults n= 1 and k = 31) from the entire set of reads [36].”. We also replaced the reference [36] with the current version of program specific to mitochondrial analysis that was used for us.

36. Hahn C, Bachmann L, Chevreux B. Reconstructing mitochondrial genomes directly from genomic next-generation sequencing reads—a baiting and iterative mapping approach, Nucleic Acids Res. 2013; 41(13), e129. doi:10.1093/nar/gkt371.

• Thank you for appointing the “BLAST spelling mistake” in line 153. It was corrected.

• We completely agree with the comment “Several another computational analysis throughout the manuscript were also not described in detail enough”. However, we worried to write a very long descriptive section without some substantial contribution to readers that can found the information from the software manual, downloading the programs and running each one according to their needs and specification. The programs used to bioinformatic analyses are often updated, so an information on a publication concerning the computational programs is very fast out of date. Thus, to improve the manuscript with some computational analysis details, we opted to include the site where the readers can download each program, the correspondent manual and where the readers can obtain more information how to run the programs. We also included the phrase “All computational analyses were performed according to software tutorials”. Moreover as possible, some details concerning the programs were also included in the manuscript.

• According the suggestion to use MUSCLE instead ClustalW to align the sequences, we performed a new alignment and did not obtained different results. However to improve the information to the readers we replaced the program in the methodology. 

48, Edgar RC. MUSCLE: multiple sequence alignment with high accuracy and high throughput. Nucleic Acids Res. 2004, 32, 1792-179. doi: 10.1093/nar/gkh340.

• We agree that in “some parts of discussion reinterred our results”, thus, we rewrote some paragraphs: excluding some information or changing the position of information to result section as requested. As instance, “The number of LAGLIDADG endonucleases encoded in intronic ncORF on rust pathogen mtDNAs ranges between three in the soybean rust (P. pachyrhizi) and nine in A. psidii.”and “ Compared with other rust pathogens, A. psidii showed a higher diversity of HEGs and intron groups: nine LAGLIDADG endonucleases, three reverse transcriptases, and six intron group IA, eight IB, two ID and two II” information were remove from discussion to result section. 

• Finally, as suggested by the reviewer 1 we speculated the role of orf208 and orf205 on A. psidii’s pathogenesis. “According to Tang et al. [87] the mitochondrial membrane protein FgLetm1, containing DUF2070, regulates mitochondrial integrity, production of endogenous reactive oxygen species and mycotoxin biosynthesis in Fusarium graminearum. The authors obtained ΔFgLetm1 mutant that showed significantly reduction on endogenous ROS levels, decreased mycotoxin deoxynivalenol biosynthesis and attenuated virulence in planta. Thus, we suggest that orf208 and orf205 encode a transmembrane protein that may be related with the fungal pathogenicity”

87. Tang G, Zhang C, Ju Z, Zheng S, Wen Z, Xu S, Chen Y, Ma Z. The mitochondrial membrane protein FgLetm1 regulates mitochondrial integrity, production of endogenous reactive oxygen species and mycotoxin biosynthesis in Fusarium graminearum. Mol. Plant Pathol. 2018, 19(7), 1595-1611. doi:10.1111/mpp.12633.

 

Response to Reviewer 2

Thank you for recognize our manuscript as well done. We appreciated all observation made by the Reviewer 2. The suggestion made certainly contributed with the correction of few mistakes as well with the improvement of our manuscript. We made clearer the novelty of our research as well as our findings and tried to explain it below. 

• We agree with the observation “The manuscript would benefit from rewriting so that one or a few main findings are put into focus that shape the storyline”. Thus, in abstract and also in other sections we rewrote and emphasized some information to make clear the novelty of our research. As instance: “A. psidii mtDNA is one of the largest rust mtDNA sequenced to date, most likely due to the abundance of ncORFs”; “Mobile genetic elements invading intron sequences may have played significant roles size but not in the shaping the structure of rust mitochondrial genome.”; “The mtDNA from rust fungi are highly syntenic.”; “Interestingly, cox1, the gene with the greatest number of introns, provided phylogenies not congruent with the core set.”; “To the best of our knowledge, this is the first report of a complete mtDNA of .A psidii, belonging to a representative of the family Sphaerophragmiacea. Our comparative mtDNA analyses also improved the knowledge concerning the biology of others rust pathogens” were included in the manuscript.

We would like to make clear to Reviewer 2 that the major contribution of our research is a detailed study of mtDNA from A. psidii and at the first time from other rust pathogens. To our knowledge none previous publication have compared the mtDNA among members of this important group of pathogen. We made a new annotation of the mtDNA of five rust pathogens, investigated the synteny inner this group, considering and making a very detailed investigation about the present of introns and intronic genes in each core gene (Table 4, S1 Table, S2 Table, S3 Table, S4 Table, S5 Table and S6 Table). 

We also would like to emphasize that A. psidii and its mtDNA has interesting aspects that deserve be detailed and discussed as made in the present research. This pathogens has a very interesting unclear information concerning its biology and its mtDNA may support clarify them, for instance, recently, taxonomic studies have led to the reclassification of A. psidii. A maximum-likelihood phylogenetic analysis using the sequences of the nuclear ribosomal RNA genes suggested that A. psidii does not belong to the genus Puccinia but instead should be within the new genus Austropuccinia of Pucciniales in the redefined family Sphaerophragmiaceae. More information about its mtDNA may contribute to validate this new classification. Moreover is the first report of a mitochondrial genome belonging to a representative of the family Sphaerophragmiacea.

Among mtDNA from rust pathogens, our research is the first one that tried to prove the presence of prediction of “hypothetical gene” from ncORFs. Using proteomic data we proved the prediction of 3, approximately 10% of ncORFs. Two of those were found exclusively in A. psidii mtDNA. We also proved the expression by RT-qPCR of one of those three, orf208 that according to literature may has function related with pathogenesis in Fusarium graminearum. This information was included in the discussion “According to Tang et al. [87] the mitochondrial membrane protein FgLetm1, containing DUF2070, regulates mitochondrial integrity, production of endogenous reactive oxygen species and mycotoxin biosynthesis in Fusarium graminearum. The authors obtained ΔFgLetm1 mutant that showed significantly reduction on endogenous ROS levels, decreased mycotoxin deoxynivalenol biosynthesis and attenuated virulence in planta. Thus, we suggest that orf208 and orf205 encode a transmembrane protein that may be related with the fungal pathogenicity”. We also highlight that a profile of ncORF from A. psidii may support the discussion of expressed genes in further studies of this pathogen.”

Finally, the observation of a non-congruence of phylogenetic analyses of 14 concatenated mtDNA core genes, 18S rDNA with cox1 is an importance contribution of our research. The recent classification used LSU-SSU barcode sequences genes, some mitochondrial genes reveled in our research may support A. psidii classification.

87. Tang G, Zhang C, Ju Z, Zheng S, Wen Z, Xu S, Chen Y, Ma Z. The mitochondrial membrane protein FgLetm1 regulates mitochondrial integrity, production of endogenous reactive oxygen species and mycotoxin biosynthesis in Fusarium graminearum. Mol. Plant Pathol. 2018, 19(7), 1595-1611. doi:10.1111/mpp.12633.

• Concerning the question, “The choice of genomes for comparison is not well explained. For example, two additional genomes from Puccinia striiformis f. sp. tritici (Pst) CY32 and P. recondita f. sp. tritici (Pt) HnZU18-3 were published last year. Why are these excluded? In particular since the former is larger than the A. psidii genome sequenced here.” we would like to inform that all analyses from our manuscript were made in 2019, unhappily before the publication Li et al. (2020). We used just the mitogenomes available during the analyses. We believe that the analyses without the mtDNA from CY32 and HnZU18-3 did not influenced our main discoveries based on the relevance of ncORFs in A. psidii mtDNA evolution, the validation of 3 “hypothetical proteins” and additional information of intronic and mobile elements in mtDNA rust pathogens. However, our manuscript was improved with additional information concerning the mtDNA from CY32 and HnZU18-3 as instance: “The mtDNA from P. striiformis f. sp. tritici ranged 102,521 [58], approximately 25% bigger than the mtDNA from P. striiformis PST-78, used in the present study” in discussion section.

58. Li C, Lu X, Zhang Y, Liu Na, Li C, Zheng W. The complete mitochondrial genomes of Puccinia striiformis f. sp. tritici and Puccinia recondita f. sp. tritici, Mitochondrial DNA B. 2020. 5(1), 29-30. doi: 10.1080/23802359.2019.1674744.

• We really appreciated the observation and question “Out of 33 ncORFs only 3 have support in proteome data, and of these only one is found at the RNA level. I would like to see more data supporting the predicted ORFs, e.g. from RNAseq or additional proteome analysis. In particular the ones with alternative start codons: could these be pseudogenes?”. We regret to inform that we did not have additional data to support the prediction of all mtDNA A. psidii ncORFs. However, the publication of ncORFs profile of A. psidii mtDNAand other rust mtDNA may be used to other research groups or us in further analyses. 

From the 33 ncORFs found by in silico analysis, 3 were validated from proteomic analyses, some remaining 30 ncDNA may be pseudogenes. However, this possibility is not strong and we appointed some information to support our view in the present letter as well explained about it in the discussion. This observation raised by the reviewer certainly made the manuscript more interesting to readers. 

Firstly, we revised the alternative start and stop codons that were found in the ncORFs from A. psidii. A new annotation was made in MFannot and revised the result manually. In first draft we found 12 ncORFs harboring alternative star or stop codons (orf118_2,, orf123_1, orf162, orf171, orf174, orf241, orf311, orf319, orf337, orf363, orf688, orf717). From the new annotation, we observed that ncORFs, orf123_1, orf162 and orf174 were wrong characterized. The correct stop codon in orf123_1 is UAG. Similar mistake were observed in orf162 and orf174 in these ncORFs the stop codon are UAG and UAA respectively. All these mistakes were corrected in the manuscript (old Table 4, currently S2 Table).

It is known that the presence of alternative codons is an indicative of pseudogenes. However, it is also known that mtDNA and other plastids genomes are composed with a great amount of genes with alternative codons, most of them related with mobile elements with functionality as the maturases (Zoschke et al. 2010, Keren et al. 2009). The 9 ncORFs (orf118_2, orf171, orf241, orf311, orf319, orf337, orf363, orf688, orf717) with alternative start codons that were found in mtDNA from A. psidii are associated with mobile elements. Moreover, it is known that most pseudogenes arise as copies of functional genes, either directly by DNA duplication or indirectly by reverse transcription of an mRNA transcript. To verify this hypothesis we blasted our sequences with the own A. psidii mtDNa and with mtDNA from other organism. None similarity was found, except with the previously annotated mobile elements. Thus, the review of ncORFs with alternative start codons support previously studies that describe alternative start codons with mobile elements related with the evolution of the mtDNA in progress (Jia and Higgs 2008, Chevalier and Stoddard, 2001). The discussion about this question was included in the discussion section “Some pseudogenes have as characteristic the presence of start or stop codons. Among the 33 ncORFs present in A. psidii mtDNA, nine harbor start alternative códons. We strong believe that they are not pseudogenes because it is also known that mtDNA and other plastids genomes are composed with a great amount of genes with alternative codons, most of them related with mobile elements with functionality as the maturases [88, 89]. The 9 ncORFs (orf118_2, orf171, orf241, orf311, orf319, orf337, orf363, orf688, orf717) with alternative start codons that were found in A. psidii mtDNA are associated with mobile elements, that may be related with the evolution of the mtDNA in progress [76, 90]. More assays using different stimuli, as well as assays carried out in planta, should be perfomerd to validate the ncORFs presence enconding protein to help the understanding the function of these ncORFs in mtDNA A. psidii”.

76. Chevalier BS, Stoddard BL. Homing endonucleases: structural and functional insight into the catalysts of intron/intein mobility. Nucleic Acids Res. 2001. 29(18): 3757–3774.doi: 10.1093/nar/29.18.375.

88. Zoschke R, Nakamura M, Liere K, Sugiura M, Börner T, Schmitz-Linneweber C. An organellar maturase associates with multiple group II introns. Natl Acad Sci U S A. 2010;107:3245–3250. doi: 10.1073/pnas.0909400107. 

89. Keren I, Bezawork-Geleta A, Kolton M, Maayan I, Belausov E, Levy M, et al. AtnMat2, a nuclear-encoded maturase required for splicing of group-II introns in Arabidopsis mitochondria. RNA. 2009;15:2299–2311. doi: 10.1261/rna.1776409. 

90. Jia W, Higgs PG, Codon usage in mitochondrial genomes: distinguishing context-dependent mutation from translational selection. Mol. Biol. Evol. 2008. 25(2). 339–351, doi: 10.1093/molbev/msm259.

• We appreciated the question “the homing endonucleases, in particular the GIY type, are highlighted. is there any evidence that they shaped evolution in this family?”, thus, we rewrote the discussion including also a paragraph about it “Many group I intron-encoded LAGLIDADG proteins also function as maturases that assist in RNA splicing [71, 72, 73, 74). This activity was described in some fungi as Saccharomyces cerevisiae, Saccharomyces capensis, Aspergillus nidulans mitochondria [72, 74, 75] suggesting that endonuclease and maturase activity to be closely coupled in both the function and evolution of LAGLIDADG proteins encoded within group I introns also in rust pathogens [76]. GIY-YIG ORFs have also been reported in introns of fungal mitochondria [76, 77]. However, any GIY-YIG endonuclease was found in rust mtDNA”. Our study is the first detailed ncORF catalogue including HEGs present in A. psidii mtDNA as well as of other rust pathogens.

71. Chi SI, Dahl M, Emblem Å, Johansen SD. Giant group I intron in a mitochondrial genome is removed by RNA back-splicing. BMC Mol Biol. 2019 20(1), 16. doi: 10.1186/s12867-019-0134-y.

72. Ho Y, Kim SJ, Waring RB. A protein encoded by a group I intron in Aspergillus nidulans directly assists RNA splicing and is a DNA endonuclease. Proc Natl Acad Sci U S A. 1997. 94(17), 8994-9. doi: 10.1073/pnas.94.17.8994. 

73. Schafer B, Wilde B, Massardo DR, Manna F, Giudice LD, Wolf K. A mitochondrial group I intron in fission yeast encodes a matures and is mobile in crosses. Curr Genet. 1994. 25, 33-341. doi: 10.1007/BF00351487.

74. Monteilhet C, Dziadkowiec D, Szczepanek T, Lazowska J. Purification and characterization of the DNA cleavage and recognition site of I-ScaI mitochondrial group I intron encoded endonuclease produced in Escherichia coli, Nucleic Acids Res. 2000. 28(5), 1245–1251. doi:10.1093/nar/28.5.1245

75. Van Ommen GJB, Boer PH, Groot GSP, de Haan M, Roosendaal E, Grivell LA, Haid A, Schweyen RJ. Mutations affecting RNA splicing and the interaction of gene expression of the yeast mitochondrial loci cob and oxi-3. Cell. 1980. 20(1), 173-183. doi:10.1016/0092-8674(80)90245-7

76. Chevalier BS, Stoddard BL. Homing endonucleases: structural and functional insight into the catalysts of intron/intein mobility. Nucleic Acids Res. 2001. 29(18): 3757–3774.doi: 10.1093/nar/29.18.375.

77. Zubaer A, Wai A, Hausner G. The fungal mitochondrial Nad5 pan-genic intron landscape, Mitochondrial DNA Part A. 2019. 30(8), 835-842. doi: 10.1080/24701394.2019.1687691.

• We improved the English in the manuscript using the edition service from American Journal Expert. The certificate is enclosed below.

• We agree with the observation “throughout the manuscript: in my opinion, it should be rust fungi, not fungi rusts.”, thus, we edited the English and rust fungi was used in whole manuscript. 

• Concerning the use of 3 different sequencing methods for a mitochondrial genome, we know that is unnecessary and unusual. We would like to explain that our initial proposal was to obtain the nuclear genome. However during our research we faced with a complex and big genome to A. psidii. Recent publications estimated 1 GB (McTaggart et al. 2018, Tobias et al. 2020). Thus, during our working in nuclear genome of A. psidii MF-1, we obtained the mtDNA of MF-1. There is not mtDNA data available to a representative member of the family Sphaerophragmiacea and also any comparative analyses of rust mtDNA. Thus, we decided that should be worthy to publish the mtDNA A.psidii analyses separately of nuclear assembled genome that is progress. 

Tobias PA, Schwessinger B, Deng CH , Wu C, Dong C, Sperschneider J, Jones A, Smith GR, Tibbits J, Chagné D, Park RF. Long read assembly of the pandemic strain of Austropuccinia psidii (myrtle rust) reveals an unusually large (gigabase sized) and repetitive fungal genome, bioRxiv preprint doi: 10.1101/2020.03.18.996108.

McTaggart AR, Duong TA, Le VQ, Shuey LS, Smidt W, Naidoo S, Wingfield MJ, Wingfield BD.Chromium sequencing: the doors open for genomics of obligate plant pathogens, BioTechniques. 2018, 65 : 253 – 257. doi: 10.2144/btn-2018-0019.

• Concerning the comment “line 236: here, I am missing an explanation of the quality parameters. What makes this assembly best?”, we agree that it was confusing and we rewrote this phrase in the manuscript “the mt genome assembly resulted in two contigs: 62,940 bp and 30,490 bp”

• We agree with the comment “line 239: When reading this, I wondered if 37 % GC is typical. Later this information is given, but this is one example that illustrates why I would like the storyline to improve.”. It is really an example of the relevance of a storyline, we rewrote all manuscript to improve it. Moreover, we included the information “very similar to other published rust mtDNA” in the results section.

• According to the suggestion made “Table 3 and table 4: Both tables are rather lengthy. I would suggest to move them to the supplements are excel files, so that the reader can filter e.g. by position. For the main text, extract the most relevant features and make corresponding tables that only show, what is also discussed in the text. Orientation for example could also be included in fig. 1”, we merged the relevant information of Table 3 and Table 4 creating a new current Table 3 and moved the old Table 3 and Table 4 to supplementary material, current S1 Table and S2 Table respectively 

• The mistakes “Line 324: orf414 should be orf 208”, “Line 326 and 330: Please adjust formatting of the reference” and “Line 356: Reference is missing” were corrected in the manuscript.

• We agree with the comment “Discussion: The more specific parts of the discussion could be moved to the results section to streamline the manuscript. In my view this would make the result section more interesting to read, and the discussion could focus on the highlight and the open questions of mitochondrial genome research.”. Thus, all discussion was rewrote, some information was moved to result section and new ones included in the discussion.

We hope to have answered all questions,

Sincerely, 

Maria Carolina Quecine

---

## [Editor Report · Decision Letter 1]

11 Feb 2021

PONE-D-20-26480R1

Revealing the high variability on nonconserved core and mobile elements of Austropuccinia psidii and other rust mitochondrial genomes

PLOS ONE

Dear Dr. Quecine,

Thank you for submitting your manuscript to PLOS ONE. After careful consideration, we feel that it has merit but does not fully meet PLOS ONE’s publication criteria as it currently stands. Therefore, we invite you to submit a revised version of the manuscript that addresses the points raised during the review process.

The following spelling/wording errors should be corrected:

1. Line 24: Please change "fungi rusts" to "rust fungi" (and make sure that this is the case throughout the text).

2. Line 104: should be "through" (not though)?

3. Line 272: Sentence is not quite clear. Should it be something like "in contrast to the conserved core genes, the ncORFs were found on both strands"?

4. Lines 329-330: M. perniciosa should be in italics.

5. Line 356: Please correct spelling mistakes in "mtDNA", "size" and "occur".

6. Please check the text again for other spelling errors etc.

We look forward to receiving your revised manuscript.

Kind regards,

Minou Nowrousian

Academic Editor

PLOS ONE

---

## [Author Response · Author response to Decision Letter 1]

17 Feb 2021

Response to Editor

We are grateful for the opportunity to resubmit again our manuscript. All mistakes were corrected in the manuscript. 

1. Line 24: Please change "fungi rusts" to "rust fungi" (and make sure that this is the case throughout the text).

R: We replaced “fungi rusts” to “rust fungi” in all manuscript.

2. Line 104: should be "through" (not though)?

R: Certainly, the correct spelling is “through”. It was corrected in the manuscript 

3. Line 272: Sentence is not quite clear. Should it be something like "in contrast to the conserved core genes, the ncORFs were found on both strands"?

R: We agree that the sentence “The ncORFs diferentlly of conserved core genes were found in both strand” was confusing and it was replaced by “In contrast to the conserved core genes, the ncORFs were found on both strands” as suggested the editor.

4. Lines 329-330: M. perniciosa should be in italics.

R: We put in italics the name M. perniciosa 

5. Line 356: Please correct spelling mistakes in "mtDNA", "size" and "occur".

6. Please check the text again for other spelling errors etc.

R: We verified carefully the whole manuscript and all mistakes were corrected.

We hope to have answered all questions,

Sincerely, 

Maria Carolina Quecine

---

## [Editor Report · Decision Letter 2]

19 Feb 2021

Revealing the high variability on nonconserved core and mobile elements of Austropuccinia psidii and other rust mitochondrial genomes

PONE-D-20-26480R2

Dear Dr. Quecine,

We’re pleased to inform you that your manuscript has been judged scientifically suitable for publication and will be formally accepted for publication once it meets all outstanding technical requirements.

Kind regards,

Minou Nowrousian

Academic Editor

PLOS ONE
---

## [Editor Report · Acceptance letter]

26 Feb 2021

PONE-D-20-26480R2 

Revealing the high variability on nonconserved core and mobile elements of *Austropuccinia psidii* and other rust mitochondrial genomes 

Dear Dr. Quecine:

I'm pleased to inform you that your manuscript has been deemed suitable for publication in PLOS ONE. Congratulations! Your manuscript is now with our production department. 

Kind regards, 

on behalf of

Dr. Minou Nowrousian 

Academic Editor

PLOS ONE